# 2021 VIVO Conference on Digital Libraries

| June 23 – 25, 2021 | ONLINE
https://vivoconference.org/vivo2021/

## Citation

Herbert, B.E., D. Lowe, D.J. Lee, and S. Potvin. 2021.  Responding to Faculty Interest in Rapid Publishing During the Pandemic: The Role of interoperable Scholarly Communication Systems at Texas A&M. 12th Annual VIVO Conference. Online. May 24-27, 2021.

## Abstract

In 2019 and 2020, the Office of Scholarly Communications pursued a strategy of the vertical integration of our scholarly communication systems in order to make them more useful to researchers, specifically our repository (DSpace), research information management system (VIVO) and Altmetrics from Digital Science.  These systems can be used to "publish" a range of documents, represent the publications on faculty Scholars@TAMU profiles, and collect engagement metrics for the publications.

We were ready, then, when faculty requests for help with special research projects while working from alternative working locations.  The faculty wanted to rapidly publish special publications that were related to the pandemic or the Black Lives Matter protests.  The outcomes from this initiative were very exciting.  Heidi Campbell edited a volume entitled *The Distanced Church: Reflections on Doing Church Online* that explored how churches worldwide were responding to the pandemic.  The volume went viral on social media, was written up in a Finnish newspaper, and was cited on a Wikipedia page.  Dr. Campbell was pleased with the experience enough to publish nine other publications through the repository, including a Spanish language version of *The Distanced Church*.  Srivi Ramasubramanian published an essay entitled *The promise and perils of interracial dialogue* in response to the BLM protests.  Again, the success of her first publication led her to curate 26 other publications in OAK Trust.  Kati Stoddard, an instructional faculty member, published an exemplary teaching resource, *Academic Honesty Quiz*, that seeks to support other faculty moving their courses online.  The resource has been downloaded almost 1000 times in the few months is has been accessible.  Finally, a community of engineering education faculty published survey results of the challenges their students faced as their classes moved online.  The teaching resource has generated more than 2000 views and a citation.  Again, the success of the project led the faculty to curate a large number of other documents in the repository.

In this talk, we will discuss the needs and interests of faculty, the role played by the library in supporting these projects, and the nature of the scholarly communication systems at Texas A&M that allow all of this to happen.
