# OpenReview forum: "Responding to Faculty Interest in Rapid Publishing During the Pandemic: The Role of interoperable Scholarly Communication Systems at Texas A&M."
_vivoconference.org/VIVO/2021/Conference_

### Official Review · Program_Chairs · 2021-06-01
**Unusual story of interest to some**

**Rating:** 6
**Confidence:** 5

**Review:**

The use of familiar systems to do unfamiliar things is potentially interesting.

The notion that local resources could be used to circumvent the traditional academic peer review publishing processes for the purpose of making "special" publications more accessible may seem bizarre to some and frightening to others.  And yet, non peer-reviewed materials are an output of universities.  This may need to be carefully explained.

The context and importance of the BLM protests to the United States may be culturally inaccessible, or difficult, for non-US audiences, as they are for some US audiences. Some context setting will be needed.  To the extent this can be done without appearing politically motivated seems daunting.

The underlying assumption here appears to be that novelty is an important aspect of scholarly communication.  Is it?